# CD4+ cell count recovery after initiation of antiretroviral therapy in HIV-infected Ethiopian adults

**Temesgen Fiseha**[1]*, **Hussen Ebrahim**[1], **Endris Ebrahim**[1], **Angesom Gebreweld**[2]

**1** Department of Clinical Laboratory Science, College of Medicine and Health Sciences, Wollo University, Dessie, Ethiopia, **2** Department of Medical Laboratory Science, College of Health Sciences, Mekelle University, Mekelle, Ethiopia

* temafiseha@gmail.com

**Data Availability Statement:** All relevant data are within the paper and its Supporting information files.

**Funding:** The author(s) received no specific funding for this work.

## Abstract

### Background

CD4+ cell count recovery after effective antiretroviral therapy (ART) is an important determinant of both AIDS and non-AIDS morbidity and mortality. Data on CD4+ cell count recovery after initiation of ART are still limited in Sub-Saharan Africa. The aim of this study was to assess CD4+ cell count recovery among HIV-infected adults initiating ART in an Ethiopian setting.

### Methods

A retrospective cohort study of HIV-infected adults initiating ART between September 2008 and June 2019 was carried out. CD4+ cell count recovery was defined as an increase in CD4+ cell count of >100 cells/mm$^3$ from baseline or achievement of a CD4+ cell count >500 cells/mm$^3$ at 12 months after ART initiation. Factors associated with CD4+ cell count recovery were evaluated using logistic regression analysis.

### Results

Of the 566 patients included in this study, the median baseline CD4+ cell count was 264 cells/mm$^3$ (IQR: 192–500). At 12 months after ART initiation, the median CD4+ cell count increased to 472 cells/mm$^3$, and the proportion of patients with CD4+ cell count < 200 cells/mm$^3$ declined from 28.3 to 15.0%. A total of 58.0% of patients had an increase in CD4+ cell count of >100 cells/mm$^3$ from baseline and 48.6% achieved a CD4+ cell count >500 cells/mm$^3$ at 12 months. Among patients with CD4+ cell counts < 200, 200–350 and >350 cells/mm$^3$ at baseline, respectively, 30%, 43.9% and 61.7% achieved a CD4+ cell count >500 cells/mm$^3$ at 12 months. In multivariable analysis, poor CD4+ cell count recovery (an increase of ≤100 cells/mm$^3$ from baseline) was associated with older age, male sex, higher baseline CD4+ cell count and zidovudine-containing initial regimen. Factors associated with poor CD4+ cell count recovery to reach the level >500 cells/mm$^3$ included older age, male sex and lower baseline CD4+ cell count.

**Competing interests:** The authors have declared that no competing interests exist.

## Conclusions

CD4+ cell count failed to recover in a substantial proportion of adults initiating ART in this resource-limited setting. Older age, male sex and baseline CD4+ cell count are the dominant factors for poor CD4+ cell count recovery. Novel therapeutic approaches are needed focusing on high risk patients to maximize CD4+ cell count recovery and improve outcomes during therapy.

## Introduction

The introduction of antiretroviral therapy (ART) among patients formerly naïve to treatment leads to suppression of HIV replication and CD4+ cell count recovery [1, 2]. Shortly after the initiation of ART, there is a rapid increase in the peripheral CD4+ cell count and CD4+ cell count recovery with ART use is associated with a significant reduction in the risk of AIDS and non-AIDS diseases or death [3–5]. Thus, CD4+ cell count recovery after initiation of ART is a potential indicator of HIV patient's clinical outcome and an increase in CD4+ cell count indicates a favorable outcome related with both AIDS and non–AIDS-related conditions and the improvement in life expectancy [4–7]. Previous studies have found CD4+ cell count increase of at least 25–50 cells/mm$^3$ during the first 12 months on ART to be correlated with improved clinical outcomes, even in the presence of detectable viremia and suggested that monitoring CD4+ cell count recovery presents an early opportunity to identify patients at risk of poorer prognosis [5, 8, 9].

Although most patients achieve CD4+ cell count recovery after effective ART, a significant proportion up to 45% do not experience an appropriate increase in their CD4+ cell counts [1, 10, 11]. Patients on ART with poor CD4+ cell count recovery, as defined by either an increase in CD4+ cell counts from baseline (e.g., $< 50$ or $< 100$ cells/mm$^3$) or a failure to achieve a CD4+ cell count over specific thresholds (e.g., 200, 350 or 500 cells/mm$^3$), are at greater risk of AIDS and serious non-AIDS morbidity and mortality [8, 12–16]. The risk of this composite outcomes associated with a poor CD4+ recovery are greater when ART was initiated at lower CD4+ cell counts [14, 17]. Several factors have been associated with poor CD4+ cell count recovery after ART initiation, including age at initiation of therapy, gender, WHO clinical disease stage, duration of untreated HIV infection, viral hepatitis coinfection, baseline CD4+ cell counts, and specific ART regimens [12, 18–21]. Genetic and environmental factors have also been linked to poor CD4+ cell count recovery during suppressive ART, even after adjustment for factors known to influence CD4+ cell count rise [12, 21–23].

Despite existing evidence that HIV-infected patients in Africa exhibit the most blunted CD4+ cell count recovery as compared with other regions–large enough to potentially influence clinical outcomes [23, 24]; data on CD4+ cell count recovery following initiation of ART are still limited in Sub-Saharan Africa where most patients initiate ART at advanced stages of disease [25, 26]. Also the factors contributing to poor CD4+ cell count recovery after initiation of ART are not well described. Such data could help to provide effective or better patient management and intervention. The aim of the present study was to assess CD4+ cell count recovery among HIV-infected adults initiating ART in an Ethiopian setting and to identify factors associated with CD4+ cell count recovery during the first 12 months of ART.

## Methods

### Study design and population

A retrospective, observational cohort study was conducted among HIV-infected adult patients initiating first-line ART at the HIV care and treatment clinic of Mehal Meda Hospital, Central Ethiopia between September 2008 and June 2019. Patients were included in this study if they received their initial first-line combination ART regimen for at least 12 months, were 18 years and older, had complete information about baseline covariates, and had CD4+ cell count results available at baseline before and 12 months after the initiation of ART. Patients with missing data for essential variables, and pregnant women were excluded from the study analysis. Ethical approval of the protocol was achieved from the Institutional Review Board of College of Medicine and Health Sciences, Wollo University. Written informed consent from patients was not required since this retrospective study only used routinely collected data, but patient records/information were anonymized and only code numbers were used throughout the study.

### Data collection and definitions

The medical records of HIV-positive patients enrolled to receive first-line ART, comprised of at least three drugs from September 2008 and June 2019 were reviewed. Baseline data including demographic (age, sex, residence, education, weight and height), clinical (WHO clinical stages, therapeutic regimens and tuberculosis), CD4+ cell count and hemoglobin level were collected. We categorized first-line regimens as either zidovudine (ZDV)–or non–ZDV-containing ART regimens. Routine viral load monitoring was not available in the sites. CD4+ cell counts are performed at baseline and every six months during follow up by FACSCount flow cytometer (Becton Dickenson and Company, California, USA) according to the manufacturer's instructions. CD4+ cell count results recorded at the baseline prior to and 12 months after the initiation of ART were taken for analysis in this study. Baseline CD4+ cell count was categorized into three categories, that is, less than 200 cells/mm$^3$, 200–350 and greater than 350 cells/mm$^3$. CD4+ cell count recovery was defined as an increase in CD4+ cell count >100 cells/mm$^3$ from baseline or achievement of an absolute CD4+ cell count threshold >500 cells/mm$^3$ at 12 months after ART initiation [12, 27–30].

### Statistical analysis

Data were entered into an "EpiData version 3.1" and analysed with SPSS version 25 software (SPSS Inc., Chicago, IL, USA). Baseline characteristics were reported as frequencies and percentages for categorical data and medians with interquartile ranges (IQR) for continuous data. Comparisons between groups were carried out using Chi-square ($x^2$) test and Mann-Whitney test, as appropriate. Wilcoxon rank sum test was used to compare median CD4+ cell counts at baseline before and 12 months after ART initiation. McNemar's test was used to compare the proportion of patients with CD4+ cell counts < 200 cells/mm$^3$ at baseline and 12 months. Changes in median CD4+ cell counts from baseline were also compared between baseline CD4+ cell count categories (<200, 201–350, and >350 cells/mm$^3$). Logistic regression analysis was used to identify factors associated with CD4+ cell count recovery. Age, sex, residence, education, body mass index (BMI), WHO clinical stages, ART regimens, CD4+ cell count, and presence of tuberculosis and anemia (hemoglobin <12.0 g/dL for women and <13.0 g/dL for men) at baseline were entered into a univariate model. Variables with *P*-values < 0.25 in the univariate analysis were included in the multivariable models using forward stepwise method. *P* values < 0.05 were considered statistically significant.

## Results

### Baseline patient characteristics

A total of 760 patients with CD4+ cell count data available at baseline met the inclusion criteria for the study. Of these, 194 did not have CD4+ cell count results at 12 months of starting ART (due to loss to follow-up, transfer-out or repeat testing not being done) and were excluded. There were no differences in the age (35 years [IQR: 28–40]), sex (54.1% female), HIV disease stage (74.7% WHO clinical stage I/II) and CD4+ cell count (261 cells/mm$^3$ [IQR: 188–562]) distribution of the excluded patients compared with patients included in the analysis (S1 Table).

Five hundred and sixty-six patients (57.2% female), with a median age at ART initiation of 36 years (IQR: 29–42) were included in this analysis. Their median bassline CD4+ cell count was 264 cells/mm$^3$ (IQR: 192–500) and 439 patients (77.6%) were at WHO clinical disease stage I/II. More than half of the patients (n = 310; 54.8%) received zidovudine (ZDV) as part of their initial ART regimen, whereas 256 (45.2%) received a non-ZDV-containing initial ART regimen. Baseline characteristics of the patients by baseline CD4+ cell count strata and overall are presented in Table 1.

### CD4+ cell count recovery after ART initiation

The median CD4+ cell count at 12 months after ART initiation was 472 cells/mm$^3$ (IQR: 294–629), with median increase from baseline of +148 cells/mm$^3$ (IQR: -2–281) ($P < 0.001$). The

**Table 1. Baseline patient characteristics by CD4+ cell count strata.**

| Characteristics | Baseline CD4+ cell count (cells/mm$^3$) | | | Total (n = 566) |
|---|---|---|---|---|
| | < 200 (n = 160) | 200–350 (n = 132) | > 350 (n = 274) | |
| Age (years), median (IQR) | 35 (28–42) | 38 (29–49) | 36 (29–41) | 36 (29–42) |
| Sex, n (%) | | | | |
| Male | 70 (43.8) | 69 (52.3) | 103 (37.6) | 242 (42.8) |
| Female | 90 (56.2) | 63 (47.7) | 171 (62.4) | 324 (57.2) |
| Residence, n (%) | | | | |
| Urban | 103 (64.4) | 74 (56.1) | 179 (65.3) | 356 (62.9) |
| Rural | 57 (35.6) | 58 (43.9) | 95 (34.7) | 210 (37.1) |
| Education, n (%) | | | | |
| < High school | 128 (80.0) | 111 (84.1) | 213 (77.7) | 452 (79.9) |
| ≥ High school | 32 (20.0) | 21 (15.9) | 61 (22.3) | 114 (20.1) |
| WHO clinical stage, n (%) | | | | |
| I/II | 113 (70.6) | 100 (75.8) | 226 (82.5) | 439 (77.6) |
| III/IV | 47 (29.4) | 32 (24.2) | 48 (17.5) | 127 (22.4) |
| Body mass index (kg/m$^2$), n (%) | | | | |
| < 18.5 | 51 (31.9) | 33 (25.0) | 76 (27.7) | 160 (28.3) |
| ≥ 18.5 | 109 (68.1) | 99 (75.0) | 198 (72.3) | 406 (71.7) |
| Tuberculosis, n (%) | 11 (6.9) | 12 (9.1) | 14 (5.1) | 37 (6.5) |
| Hemoglobin (g/dl), median (IQR) | 12.1 (9.7–15.0) | 13.7 (12.1–15.0) | 13.4 (11.9–15.6) | 12.6 (11.4–14.7) |
| Initial regimen, n (%) | | | | |
| AZT-3TC-NVP | 66 (41.8) | 36 (27.3) | 112 (40.9) | 214 (37.8) |
| AZT-3TC-EFV | 32 (20.0) | 24 (18.2) | 40 (14.6) | 96 (17.0) |
| TDF-3TC-EFV | 42 (26.3) | 50 (37.8) | 81 (29.6) | 173 (30.6) |
| TDF-3TC-NVP | 12 (7.5) | 20 (15.2) | 37 (13.5) | 69 (12.2) |
| ABC-3TC-EFV/NVP | 8 (5.0) | 2 (1.5) | 4 (1.4) | 14 (2.4) |

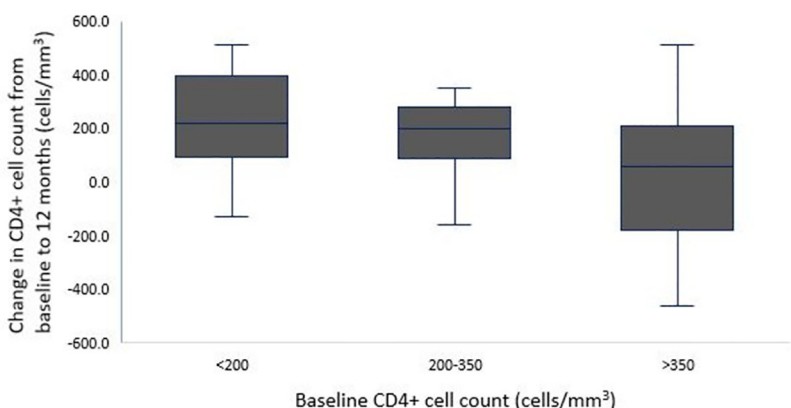

**Fig 1. Median increase in CD4+ cell count from baseline to 12 months by baseline CD4+ cell count.**

proportion of patients with CD4+ cell count < 200 cells/mm³ declined from 28.3% at baseline to 14.3% at 12 months ($P < 0.001$). The median CD4+ cell count increased from 152 cells/mm³ (IQR: 111–175) at baseline to 388 cells/mm³ (IQR: 216–561) at 12 months in patients with a baseline CD4+ cell count < 200 cells/mm³ ($P < 0.001$), from 238 (IQR: 228–249) to 445 cells/mm³ (IQR: 334–519) in patients with baseline CD4+ cell counts of 200–350 cells/mm³ ($P < 0.001$), and from 502 (IQR: 426–823) to 581 cells/mm³ (IQR: 333–701) in patients with a baseline CD4+ cell count >350 cells/mm³ ($P = 0.806$). Median CD4+ cell count increases from baseline were +220 (IQR: 94–399) and +199 cells/mm³ (IQR: 87–281) in the patients with baseline CD4+ cell counts of < 200 and 200–350 cells/mm³, respectively compared to only +60 (IQR: -185–212) in those with CD4+ counts >350 cells/mm³ ($P < 0.001$) (Fig 1).

A total of 58.0% (95% CI 53.9–62.0%) patients had an increase in CD4+ cell count of >100 cells/mm³ from baseline and 48.6% (95% CI 44.5–52.7%) achieved an absolute CD4+ cell count >500 cells/mm³ at 12 months after initiating ART. In addition, only 14.0% (95% CI 11.3–17.0%) of the patients were able to achieve the median reference CD4+ cell threshold counts in Ethiopians (~760 cells/mm³) [31]. The proportion of patients who achieved a CD4+ cell count of >500 cells/mm³ after receiving 12 months of suppressive ART were 30.0%, 43.9%, and 61.7% for the patients who initiated ART with CD4+ cell counts of < 200, 200–350, and > 350 cells/mm³, respectively ($P < 0.001$). The proportion of patients who achieved a CD4+ cell count of 760 cells/mm³ after receiving 12 months of suppressive ART were 6.9%, 10.6%, and 19.7% for those who initiated ART with a CD4+ cell count < 200, 200–350, and > 350 cells/mm³, respectively ($P = 0.011$).

### Factors associated with CD4+ cell count recovery

In a univariate analysis, the factors found to be associated with poor CD4+ cell count recovery (an increase of ≤ 100 cells/mm³ from baseline) at 12 months of ART were older age (COR = 1.94, 95% CI 1.37–2.74), male sex (COR = 1.82, 95% CI 1.30–2.56), CD4+ cell count > 350 cells/mm³ (COR = 3.33, 95% CI 2.19–5.06), anemia (COR = 0.66, 95% CI 0.47–0.94) and ZDV-containing initial regimen (COR = 1.59, 95% CI 1.13–2.23). Older age (AOR = 2.07, 95% CI 1.41–3.03), male sex (AOR = 1.93, 95% CI 1.32–2.81), high baseline CD4+ cell count (AOR = 4.01, 95% CI 2.57–6.255) and ZDV-containing initial regimen (AOR = 1.60, 95% CI

**Table 2. Factors associated with poor CD4+ cell count recovery (an increase of $\leq 100$ cells/mm$^3$ from baseline) at 12 months.**

| Variables | Crude OR (95% CI) | *P*-value | Adjusted OR (95% CI) | *P*-value |
|---|---|---|---|---|
| Age (years) | | < 0.001 | | < 0.001 |
| > 40 | 1.94 (1.37–2.74) | | 2.07 (1.41–3.03) | |
| ≤ 40 | 1 | | 1 | |
| Sex | | < 0.001 | | 0.001 |
| Male | 1.82 (1.30–2.56) | | 1.93 (1.32–2.81) | |
| Female | 1 | | 1 | |
| Residence | | 0.448 | | |
| Urban | 1.14 (0.81–1.62) | | | |
| Rural | 1 | | | |
| Educational level | | 0.198 | | 0.506 |
| < High school | 1.35 (0.87–1.96) | | 0.86 (0.54–1.35) | |
| ≥ High school | 1 | | 1 | |
| WHO clinical stage | | 0.348 | | |
| Stage III/IV | 1.21 (0.81–1.80) | | | |
| Stage I/II | 1 | | | |
| Baseline CD4+ cell count (cells/mm$^3$) | | | | |
| < 200 | 1 | | 1 | |
| 200–350 | 1.03 (0.62–1.72) | 0.901 | 0.94 (0.57–1.67) | 0.922 |
| > 350 | 3.33 (2.19–5.06) | < 0.001 | 4.01 (2.57–6.25) | < 0.001 |
| Body mass index (kg/m$^2$) | | 0.892 | | |
| < 18.5 | 1.03 (0.71–1.49) | | | |
| ≥ 18.5 | 1 | | | |
| Tuberculosis status | | 0.061 | | 0.052 |
| Yes | 1.89 (0.96–3.70) | | 2.10 (0.99–4.45) | |
| No | 1 | | 1 | |
| Anemia status | | 0.021 | | 0.075 |
| Yes | 0.66 (0.47–0.94) | | 0.70 (0.48–1.04) | |
| No | 1 | | 1 | |
| ART regimen | | 0.007 | | 0.012 |
| ZDV-containing | 1.59 (1.13–2.23) | | 1.60 (1.11–2.32) | |
| Non-ZDV containing | 1 | | 1 | |

1.11–2.32) remained significantly associated with poor CD4+ cell count recovery in the multivariable analysis (Table 2).

Among the 434 patients who initiated ART with CD4+ cell counts $\leq 500$ cells/mm$^3$, 54.1% (95% CI 49.4–58.8%) failed to recover their CD4+ cell count to >500 cells/mm$^3$ at 12 months. By univariate analyses, older age (COR = 2.07, 95% CI 1.40–3.07), male sex (COR = 2.57, 95% CI 1.69–3.75), WHO clinical stage III/IV (COR = 1.88, 95% CI 1.18–3.02), baseline CD4+ cell counts < 200 cells/mm$^3$ (COR = 4.43, 95% CI 2.73–7.18) and 200–350 cells/mm$^3$ (COR = 2.42, 95% CI 1.49–3.94), and presence of tuberculosis (COR = 1.34, 95% CI 1.05–1.71) were associated with poor CD4+ cell count recovery to >500 cells/mm$^3$. The multivariable analysis revealed that older age (AOR = 1.99, 95% CI 1.28–3.08), male sex (AOR = 2.15, 95% CI 1.39–3.33), and baseline CD4+ cell counts < 200 cells/mm$^3$ (AOR = 4.45, 95% CI 2.67–7.41) and 200–350 cells/mm$^3$ (AOR = 2.03, 95% CI 1.21–3.39) were independently associated with poor CD4+ cell count recovery to >500 cells/mm$^3$ at 12 months (Table 3).

**Table 3. Factors associated with poor CD4+ cell count recovery to >500 cells/mm³ among 434 patients initiating ART with CD4+ cell counts ≤ 500 cells/mm³ at 12 months.**

| Variables | Crude OR (95% CI) | P-value | Adjusted OR (95% CI) | P-value |
|---|---|---|---|---|
| Age (years) | | < 0.001 | | 0.002 |
| > 40 | 2.07 (1.40–3.07) | | 1.99 (1.28–3.08) | |
| ≤ 40 | 1 | | 1 | |
| Sex | | < 0.001 | | 0.001 |
| Male | 2.52 (1.69–3.75) | | 2.15 (1.39–3.33) | |
| Female | 1 | | 1 | |
| Residence | | 0.623 | | |
| Urban | 0.95 (0.62–1.34) | | | |
| Rural | 1 | | | |
| Educational level | | 0.652 | | |
| < High school | 1.12 (0.69–1.88) | | | |
| ≥ High school | 1 | | | |
| WHO clinical stage | | 0.008 | | 0.458 |
| Stage III/IV | 1.88 (1.18–3.02) | | 1.23 (0.71–2.11) | |
| Stage I/II | 1 | | | |
| Baseline CD4+ cell count (cells/mm³) | | | | 0.039 |
| < 200 | 4.43 (2.73–7.18) | < 0.001 | 4.45 (2.67–7.41) | < 0.001 |
| 200–350 | 2.42 (1.49–3.94) | < 0.001 | 2.03 (1.21–3.39) | 0.007 |
| > 350 | 1 | | | |
| Body mass index (kg/m²) | | 0.884 | | |
| < 18.5 | 1.03 (0.68–1.56) | | | |
| ≥ 18.5 | 1 | | | |
| Tuberculosis status | | 0.047 | | 0.071 |
| Yes | 1.34 (1.05–1.71) | | 2.24 (0.93–5.39) | |
| No | 1 | | 1 | |
| Anemia status | | 0.754 | | |
| Yes | 0.94 (0.64–1.38) | | | |
| No | 1 | | | |
| ART regimen | | 0.191 | | 0.655 |
| ZDV-containing | 1.29 (0.88–1.88) | | 0.91 (0.59–1.39) | |
| Non-ZDV containing | 1 | | 1 | |

## Discussion

In this study, there was a significant increase in CD4+ cell count and a decrease in the proportion of patients with severe immunosuppression during the first 12 months after ART initiation. Larger increases in CD4+ cell counts were observed in patients who started ART with lower CD4+ cell counts. A substantial proportion patients failed to recover their CD4+ cell count 12 months after initiation of therapy. Older age, male sex and CD4+ cell counts at the time of ART initiation were the major factors associated with poor CD4+ cell count recovery.

The median baseline CD4+ cell count of this study (264 cells/mm³) was comparable to some of the studies in the region, which reported median CD4+ cell counts of 240 cells/mm³ [32] and 257 cells/mm³ [27] at ART initiation. This was, however, higher than the median baseline CD4+ cell counts reported in other African studies, including 152 and 201 cells/mm³ in Northern Ethiopia [28, 29], 144 cells/mm³ in Northwest Ethiopia [30], 142 cells/mm³ in Nigeria [33] and 147 cells/mm³ in six sub-Saharan African countries [34]. With regard to the

proportion of patients initiating ART late (CD4+ cell counts < 200 cells/mm$^3$), the current study was similar to the Mongolian study [35] where 24.7% of patients initiated with CD4+ counts < 200 cells/mm$^3$. Other studies in the region reported a higher proportion, up to 76.8% of patients starting ART late [27–29, 33, 34, 36]. Our findings are very encouraging for the achievement of the ambitious UNAIDS 90-90-90 targets [37].

At 12 months after initiation of ART, the median CD4+ cell count increased to 472 cells/ mm$^3$ (an increase of +148 cells/mm$^3$ from baseline) and the proportion of patients with CD4+ cell counts < 200 cells/mm$^3$ decreased from 28.3 to 15.0%. This supports data from other studies that ART can led to an increase in CD4+ cell counts and a decrease in the proportion of patients with severe immunosuppression [18, 29, 30, 33, 38–40]. In a study from the Ethiopian HIV cohort [29], the median increase in CD4+ count after ART was from 201 to 423 cells/ mm$^3$, and the proportion of patients with CD4+ count < 200 cells/mm$^3$ decreased from 49.6 to 15.6%. In another Ethiopian HIV cohort study [30], the median CD4+ cell count increased from 144 cells/mm$^3$ at baseline to 266 cells/mm$^3$ at 12 months, and the proportion of patients with CD4+ count < 100 cells/mm$^3$ decreased from 31 to 6%. In the South African HIV cohort study [40], the median CD4$^+$ cell count increased from 97 to 261 cells/mm$^3$ at 48 weeks and the proportion of patients with CD4+ count < 100 cells/mm$^3$ decrease from 51 to 4%. In this study, the increase in CD4+ cell count varied according to baseline CD4+ counts and was larger in patients with low counts compared to those with high counts. Our results are similar to reports from other studies, indicating that a low baseline CD4+ count does not preclude an excellent CD4+ cell count response to ART [40–42]. This finding is clinically important, because a higher CD4+ cell count is associated with the greatest benefit for patients on ART with a low CD4+ count [3].

Our study demonstrated that 58% of patients had an increase in CD4+ cell count of >100 cells/mm$^3$ from baseline at 12 months after therapy; a result in agreement with numerous other studies [11, 14, 39, 43]. In an urban HIV cohort in Uganda [11], for example, 55% of patients had a CD4+ cell count increase of > 100 cells/mm$^3$ at 12 months. In the COHERE collaboration cohort study [43], 59.2% of patients experienced a CD4+ cell count increase of > 100 cells/mm$^3$ at 12 months. In a Rwandan HIV cohort [32], 70.0% of patients had an increase in CD4+ cell count ≥100 cells/mm$^3$ at 12 months. In the Spain HIV cohort study [39], 73.2% of patients had a CD4+ cell count increase of > 100 cells/mm$^3$ at 12 months. Similar to other studies, we found that older age at ART initiation [20, 32, 40, 43], male sex [20, 28], and higher baseline CD4+ cell count [11, 28, 40, 43] are associated with poor CD4+ cell count recovery, defined as a CD4+ cell count increase of ≤ 100 cells/mm$^3$ from baseline at 12 months. We also found that patients who initiated on ZDV-containing ART were more likely to have poor CD4+ cell count recovery than patients on non-ZDV-containing ART [11, 20].

About 49% of our patients reached a CD4+ cell count >500 cells/mm$^3$ at 12 months of ART. Of note, patients who regain their CD4+ cell count to this immunological point have a better clinical outcome with both HIV- and non-HIV-related morbidity and mortality [13, 15]. Studies from other parts of the country have estimated that 37.6% [36] and 38.8% [29] of patients had reached CD4+ cell counts >500 cells/mm$^3$ after ART start. Recently, one Ethiopian HIV cohort study reported 39% of patients reached a CD4+ cell count >500 cells/mm$^3$ at 12 months [27]. The South African HIV cohort study reported 6.8% of patients achieved a CD4+ cell count >500 cells/mm$^3$ at 48 weeks [40]. Nearly 62% of our patients initiated ART with CD4+ counts >350 cells/mm$^3$ achieved a CD4+ cell count >500 cells/mm$^3$, while 70% of patients with CD4+ counts < 200 cells/mm$^3$ did not. Studies have reported that individuals initiating ART at higher counts have their CD4+ cell count return to nearly normal or normal (>500 cells/mm$^3$) than those who initiated at lower counts (< 200 cells/mm$^3$) [18, 28, 42, 44].

These findings add to the evidence suggesting that, to facilitate immune recovery, ART should be started before CD4+ count has fallen below 200 cells/mm$^3$.

We found that factors including older age, male sex and low CD4+ cell counts at baseline were associated with CD4+ cell count recovery to >500 cells/mm$^3$. Older age is a well-recognized risk factor for poor CD4+ cell count recovery after receipt of ART [1, 18, 34, 45–47]. Since thymic function decreases with aging, patient age at initiation could influence CD4+ cell recovery [48]. A study from Northwest Ethiopia illustrated that the CD4+ cell count decrease by 5.0 cells/mm$^3$ for each additional 1 year of baseline age [36]. The sub-Saharan Africa cohort study showed that older patients had a significantly longer time to, and lower rate of, achieving a CD4+ cell count >500 cells/mm$^3$ [49]. This study reported that the delay in achieving a robust immune response could have significant implications for the risk of comorbidities associated with age. Consistent with previous studies, male sex was associated with poor CD4+ cell count recovery to >500 cells/mm$^3$ [34, 45, 50]. Only 37.6% meals in this study initiated ART with CD4+ counts >350 cells/mm$^3$ compared with 62.4% females. Females may normally have higher CD4+ cell counts than males [31, 51] and some studies also underlined the effect of male hormones on the thymic function [52].

Our results are consistent with those from prior studies suggesting that CD4+ cell count recovery after ART depends heavily on the baseline levels with patients starting with low CD4+ counts failing to recover CD4+ cell count to >500 cells/mm$^3$ [10, 18, 34, 35, 46, 47]. In the Italian HIV cohort study, having baseline CD4+ count ≤ 350 cells/mm$^3$ was associated with poor CD4+ cell count recovery to >500/mm$^3$ [46]. The Johns Hopkins HIV cohort study reported that waiting to start ART at low CD4+ count (≤ 350 cells/mm$^3$) was associated with failure to recover CD4+ cell count to >500/mm$^3$ [53]. The FHDH HIV cohort study reported that a higher CD4+ count at ART initiation was strongly associated with a higher probability of CD4+ cell count recovery to >500 cells/mm$^3$ [54]. Other studies even reported that recovery to CD4+ cell count >500 cells/mm$^3$ may be attainable only in patients starting with counts >350 cells/mm$^3$ [55, 56]. These results support current guidelines to start ART in all patients before they reach a critical CD4+ cell count and suggest that there may be immunological benefits associated with initiating therapy at even higher CD4+ counts.

Our study has several limitations; the first is the observational design, subject to the possible effects of confounders. To be included in the present analysis, patients must have had a CD4+ cell count results at baseline and 12 months after initiating ART. By excluding some patients who died or were lost to follow-up during the first 12 months of enrollment, CD4+ cell count recovery associated with ART may have been overestimated; however, patients without a follow-up CD4+ cell count results at 12 months had similar baseline characteristics as the patients in our analysis. The role of baseline cumulative viremia on CD4+ cell count recovery has not been studied. Lastly, we did not take into account adherence to ART and other factors such as alcohol consumption and mental health status.

## Conclusions

In conclusion, CD4+ cell count failed to recover in a substantial proportion of patients initiating ART in this resource-limited setting. Older age, male sex and CD4+ cell count at the initiation of ART are the dominant factors for poor CD4+ cell count recovery. In addition, patients initiating zidovudine-containing ART regimen should be identified as groups at higher risk for poor CD4+ cell count recovery. Therefore, novel therapeutic approaches, with good access to CD4+ cell count monitoring and a focus on those at greatest risk, are needed to maximize CD4+ cell count recovery and improve outcomes during therapy.

## Supporting information

**S1 Dataset. The excel database used for this manuscript.**
(XLSX)

**S1 Table. Baseline characteristics of patients included and excluded from the analysis.**
(DOCX)

## Acknowledgments

The authors acknowledge the health staff at the Mehal Meda Hospital HIV care and treatment clinic for their assistance in gathering the data.

## Author Contributions

**Conceptualization:** Temesgen Fiseha, Endris Ebrahim, Angesom Gebreweld.

**Data curation:** Temesgen Fiseha, Hussen Ebrahim, Angesom Gebreweld.

**Formal analysis:** Temesgen Fiseha, Hussen Ebrahim, Endris Ebrahim.

**Methodology:** Temesgen Fiseha, Hussen Ebrahim, Endris Ebrahim, Angesom Gebreweld.

**Software:** Temesgen Fiseha.

**Visualization:** Hussen Ebrahim, Endris Ebrahim, Angesom Gebreweld.

**Writing – original draft:** Temesgen Fiseha.

**Writing – review & editing:** Temesgen Fiseha, Hussen Ebrahim, Endris Ebrahim, Angesom Gebreweld.

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
