## [Decision Letter · Decision Letter 0]

6 Jan 2022

PONE-D-21-27285CD4+ cell count recovery after initiation of antiretroviral therapy in HIV-infected Ethiopian adultsPLOS ONE

Dear Dr. Fiseha,

Thank you for submitting your manuscript to PLOS ONE. After careful consideration, we feel that it has merit but does not fully meet PLOS ONE’s publication criteria as it currently stands. Therefore, we invite you to submit a revised version of the manuscript that addresses the points raised during the review process.

We look forward to receiving your revised manuscript.

Kind regards,

Ngai Sze Wong

Academic Editor

PLOS ONE

Journal Requirements:

3. Please upload a copy of Supporting Information which you refer to in your text on page 14.

Additional Editor Comments:

Two additional very minor points for revision:

-Abstract: typo in RESULT of CD4 cell count >5000. It should be 500 instead.

-The outcome variable name for "poor CD4+ cell count recovery to >500 cells/mm3" was a bit difficult to understand without looking at the details. Did you mean poor recovery to reach the level above 500?

Reviewers' comments:

Reviewer's Responses to Questions

**Comments to the Author**

1. Is the manuscript technically sound, and do the data support the conclusions?

Reviewer #1: Yes

2. Has the statistical analysis been performed appropriately and rigorously? 

Reviewer #1: I Don't Know

3. Have the authors made all data underlying the findings in their manuscript fully available?

Reviewer #1: Yes

4. Is the manuscript presented in an intelligible fashion and written in standard English?

Reviewer #1: Yes

5. Review Comments to the Author

Reviewer #1: This work documents the degree to which CD4 recovery occurs in an reasonably large Ethiopian cohort during the first 12 months of HIV therapy. The findings are consistent with those from other cohorts and reveals particularly for those patients starting therapy at low CD4 counts recovery is suboptimal. This is not uncommon at older ages, and in males but can occur for a large percentage of the population. While this is consistent across the world, there is not as much data from Africa, where the data is most relevant.

Since they appear to have the possibility of long-term follow-up what would substantially raise the value of this work would be some effort to document whether or not there were clinical consequences of either a <100 cell rise in CD4 counts, and/ or CD4 counts staying below 500 or some other lower thresholds. One Ugandan study they cited (Nakanjako et. Al. 2008) found no clinical difference between suboptimal and “more optimal” CD4 recovery, but the length of follow up was not as long as this cohort has. Any clinical data including mortality, hospitalization, tuberculosis acquisition/diagnosis would strengthen the emerging data that something in addition to antiviral therapy is needed in patients with suboptimal therapy.

Please also respond to comment in attached manuscript.

6. PLOS authors have the option to publish the peer review history of their article (what does this mean?). If published, this will include your full peer review and any attached files.

Reviewer #1: No

---

## [Author Response · Author response to Decision Letter 0]

17 Feb 2022

Response to Journal Requirements

Comment # 1: Please ensure that your manuscript meets PLOS ONE's style requirements, including those for file naming. The PLOS ONE style templates can be found at

Comment # 2: We note that you have included the phrase “data not shown” in your manuscript. Unfortunately, this does not meet our data sharing requirements. PLOS does not permit references to inaccessible data. We require that authors provide all relevant data within the paper, Supporting Information files, or in an acceptable, public repository. Please add a citation to support this phrase or upload the data that corresponds with these findings to a stable repository (such as Figshare or Dryad) and provide and URLs, DOIs, or accession numbers that may be used to access these data. Or, if the data are not a core part of the research being presented in your study, we ask that you remove the phrase that refers to these data.

Response #2: As suggested, it is provided within the paper as Supporting Information file “S1 Table. Baseline characteristics of patients included and excluded from the analysis. (DOCX)” (Line 315-316), and cited within the main text as “S1 Table” (Line 138)

Comment # 3: Please upload a copy of Supporting Information which you refer to in your text on page 14.

Response #3: As suggested, a copy of Supporting Information was uploaded

Comment # 4: Please review your reference list to ensure that it is complete and correct. If you have cited papers that have been retracted, please include the rationale for doing so in the manuscript text, or remove these references and replace them with relevant current references. Any changes to the reference list should be mentioned in the rebuttal letter that accompanies your revised manuscript. If you need to cite a retracted article, indicate the article’s retracted status in the References list and also include a citation and full reference for the retraction notice.

Response #4: As suggested, we reviewed our reference list to ensure that it is complete and correct, and the following corrections were made: 

Reference no. 

1. “…. Lancet Infect Dis. 2006 May 1;6(5):280–7” is stated as “…. Lancet Infect Dis. 2006;6(5):280–7” as 

4. “… AIDS. 2008 Apr 23;22(7):841–848.” is stated as “…. AIDS. 2008; 22(7):841–8”

7. “… AIDS. 2014;28:1193–202 .” is stated as “…. AIDS. 2014;28(8):1193–202.”

10. “… Arch Intern Med. 2003 Oct 13;163(18):2187–95” is stated as “…. Arch Intern Med. 2003;163(18):2187–95”

14. “… AIDS and Non-AIDS Diseases. JAIDS J Acquir Immune Defic Syndr. 2008 Aug 15;48(5):541–546” is stated as “…. AIDS and Non-AIDS Diseases. J Acquir Immune Defic Syndr. 2008;48(5):541–6”

22. “…. J Infect Dis. 2006;194:1098–1107” is stated as “…. J Infect Dis. 2006;194 (8):1098–107”

25. “… J Acquir Immune Defic Syndr. 2009;52(2):280” is stated as “…. J Acquir Immune Defic Syndr. 2009;52(2):280-9”

27. “… Cross-Sectional Study. HIV AIDS. 12:69–77” is stated as “…. Cross-Sectional Study. HIV AIDS. 2020; 12:69–77”

29. “… PLOS ONE. 2019 Dec 12;14(12):e0226293” is stated as “…. PLOS ONE. 2019;14(12):e0226293”

30. “… BMC Infect Dis. 2014 Jan 14;14(1):28” is stated as “…. BMC Infect Dis. 2014;14(1):28”

34. “… AIDS. 2018 May 15;32(8):1043–1051” is stated as “…. AIDS. 2018;32(8):1043–51”

42. “… Clin Infect Dis. 2009 Mar 15;48(6):787–94” is stated as “…. Clin Infect Dis. 2009;48(6):787–94”

49. “… AIDS. 2018 Jan 2;32(1):25–34” is stated as “…. AIDS. 2018;32(1):25–34”

Response to Additional Editor Comments

Two additional very minor points for revision:

Comment # 1: Abstract: typo in RESULT of CD4 cell count >5000. It should be 500 instead.

Response #1: As suggested, it is stated as “>500” (Line 43)

Comment # 2: The outcome variable name for "poor CD4+ cell count recovery to >500 cells/mm3" was a bit difficult to understand without looking at the details. Did you mean poor recovery to reach the level above 500?

Response #2: Yes, it is to mean poor recovery to reach the level above 500. As suggested, it is stated as “poor CD4+ cell count recovery to reach the level >500” (Line 46-47)

Response to Reviewer Comments 

Reviewer #1 

This work documents the degree to which CD4 recovery occurs in an reasonably large Ethiopian cohort during the first 12 months of HIV therapy. The findings are consistent with those from other cohorts and reveals particularly for those patients starting therapy at low CD4 counts recovery is suboptimal. This is not uncommon at older ages, and in males but can occur for a large percentage of the population. While this is consistent across the world, there is not as much data from Africa, where the data is most relevant.

Comment # 1: Since they appear to have the possibility of long-term follow-up what would substantially raise the value of this work would be some effort to document whether or not there were clinical consequences of either a <100 cell rise in CD4 counts, and/ or CD4 counts staying below 500 or some other lower thresholds. One Ugandan study they cited (Nakanjako et. Al. 2008) found no clinical difference between suboptimal and “more optimal” CD4 recovery, but the length of follow up was not as long as this cohort has. Any clinical data including mortality, hospitalization, tuberculosis acquisition/diagnosis would strengthen the emerging data that something in addition to antiviral therapy is needed in patients with suboptimal therapy.

Response #1: Our objective was to study the frequency and potential determinants of CD4+ cell count recovery, and we did not evaluate the clinical relevance of either a <100 cell rise in CD4 counts, and/ or CD4 counts staying below 500 or some other lower thresholds. Yes, but such clinical data from the clinical files was not documented.

Comment # 2: Blunted compared to what cohorts

Response #2: As suggested, it is stated as “as compared with other regions” (Line 79)

---

## [Editor Report · Decision Letter 1]

8 Mar 2022

CD4+ cell count recovery after initiation of antiretroviral therapy in HIV-infected Ethiopian adults

PONE-D-21-27285R1

Dear Dr. Fiseha,

We’re pleased to inform you that your manuscript has been judged scientifically suitable for publication and will be formally accepted for publication once it meets all outstanding technical requirements.

Kind regards,

Ngai Sze Wong

Academic Editor

PLOS ONE

---

## [Editor Report · Acceptance letter]

15 Mar 2022

PONE-D-21-27285R1 

CD4+ cell count recovery after initiation of antiretroviral therapy in HIV-infected Ethiopian adults 

Dear Dr. Fiseha:

I'm pleased to inform you that your manuscript has been deemed suitable for publication in PLOS ONE. Congratulations! Your manuscript is now with our production department. 

Kind regards, 

on behalf of

Dr. Ngai Sze Wong 

Academic Editor

PLOS ONE